# Drosophila medulla neuroblast termination via apoptosis, differentiation, and gliogenic switch is scheduled by the depletion of the neuroepithelial stem cell pool

**Phuong-Khanh Nguyen[1,2], Louise Y Cheng[1,2,3]***

[1]Peter MacCallum Cancer Centre, Melbourne, Australia; [2]Department of Anatomy and Physiology, The University of Melbourne, Melbourne, Australia; [3]Sir Peter MacCallum Department of Oncology, The University of Melbourne, Melbourne, Australia

**Abstract** The brain is consisted of diverse neurons arising from a limited number of neural stem cells. *Drosophila* neural stem cells called neuroblasts (NBs) produces specific neural lineages of various lineage sizes depending on their location in the brain. In the *Drosophila* visual processing centre - the optic lobes (OLs), medulla NBs derived from the neuroepithelium (NE) give rise to neurons and glia cells of the medulla cortex. The timing and the mechanisms responsible for the cessation of medulla NBs are so far not known. In this study, we show that the termination of medulla NBs during early pupal development is determined by the exhaustion of the NE stem cell pool. Hence, altering NE-NB transition during larval neurogenesis disrupts the timely termination of medulla NBs. Medulla NBs terminate neurogenesis via a combination of apoptosis, terminal symmetric division via Prospero, and a switch to gliogenesis via Glial Cell Missing (Gcm); however, these processes occur independently of each other. We also show that temporal progression of the medulla NBs is mostly not required for their termination. As the *Drosophila* OL shares a similar mode of division with mammalian neurogenesis, understanding when and how these progenitors cease proliferation during development can have important implications for mammalian brain size determination and regulation of its overall function.

***For correspondence:**
louise.cheng@petermac.org

**Competing interest:** The authors declare that no competing interests exist.

## Editor's evaluation

This study presents an important finding on the mechanisms by which medulla neural stem cells in *Drosophila* optic lobe, whose division mode largely resembles mammalian neural stem cells, terminate cell proliferation in neural development. The evidence supporting the conclusions of the authors is convincing. The work will be of interest to stem cell biologists and developmental biologists.

## Introduction

The CNS is the cognitive control centre of the body and is generated by neural stem cells (NSCs) during development. *Drosophila* NSCs, called neuroblasts (NBs), produce different neural and glial cell types and subtypes that contribute to the formation of the adult brain. The cellular diversity of the CNS is determined in a region-specific manner, via differential control of NB division mode and their

**eLife digest** Every cell in the body can be traced back to a stem cell. For instance, most cells in the adult brains of fruit flies come from a type of stem cell known as a neuroblast. This includes neurons and glial cells (which support and protect neurons) in the optic lobe, the part of the brain that processes visual information.

The numbers of neurons and glia in the optic lobe are tightly regulated such that when the right numbers are reached, the neuroblasts stop making more and are terminated. But how and when this occurs is poorly understood.

To investigate, Nguyen and Cheng studied when neuroblasts disappear in the optic lobe over the course of development. This revealed that the number of neuroblasts dropped drastically 12 to 18 hours after the fruit fly larvae developed in to pupae, and were completely gone by 30 hours in to pupae life.

Further experiments revealed that the timing of this decrease is influenced by neuroepithelium cells, the pool of stem cells that generate neuroblasts during the early stages of development. Nguyen and Cheng found that speeding up this transition so that neuroblasts arise from the neuroepithelium earlier, led neuroblasts to disappear faster from the optic lobe; whereas delaying the transition caused neuroblasts to persist for much longer. Thus, the time at which neuroblasts are born determines when they are terminated.

Furthermore, Nguyen and Cheng showed that the neuroblasts were lost through a combination of means. This includes dying via a process called apoptosis, dividing to form two mature neurons, or switching to a glial cell fate.

These findings provide a deeper understanding of the mechanisms regulating stem cell pools and their conversion to different cell types, a process that is crucial to the proper development of the brain. How cells divide to form the optic lobe of fruit flies is similar to how new neurons arise in the mammalian brain. Understanding how and when stem cells in the fruit fly brain stop proliferating could therefore provide new insights in to the development of the human brain.

temporal identity (reviewed by *Harding and White, 2018*). The majority of NBs are Type I NBs located in the ventral nerve cord (VNC) and the central brain (CB). They divide asymmetrically to generate a NB and a smaller ganglion mother cell (GMC) that undergoes terminal differentiation to give rise to neurons or glial cells (*Harding and White, 2018*). In addition, in each brain hemisphere, there are eight Type II NBs located on the dorsal surface of the CB, which asymmetrically divide to give rise to a NB and an intermediate neural progenitor (INP; *Bello et al., 2008*; *Boone and Doe, 2008*; *Bowman et al., 2008*). INPs undergo maturation prior to additional rounds of asymmetric division, to give rise to neurons or glial cells.

The optic lobes (OLs) are the visual processing centre of the adult fly brain (*Figure 1A*). During neural development, neurogenesis in the OLs takes place in two proliferation centres: the inner and the outer proliferation centres (OPC), which produce different neural cell types that populate various cortexes of the visual system (*Apitz and Salecker, 2014*; *Sato et al., 2019*). In the OPC, medulla NBs are generated from a pseudostratified neuroepithelium through a differentiating 'proneural wave' (*Figure 1A'*; *Egger et al., 2007*; *Li et al., 2013*; *Yasugi et al., 2008*). This proneural wave is characterised by the expression of proneural factors such as Lethal of scute (L'sc) which precedes NB generation from NE cells (*Yasugi et al., 2008*). During early larval stages, OPC NE cells undergo symmetric cell divisions to expand the progenitor population. By early-third instar, progression of the proneural wave is initiated in the NE cells at the medial edge, leading to their differentiation into medulla NBs. Medulla NBs sequentially express a series of temporal transcription factors according to their birth order (*Konstantinides et al., 2022*; *Li et al., 2013*; *Zhu et al., 2022*). During the early-mid temporal stages, NBs divide asymmetrically to self-renew and to generate a GMC that in turn produces two neurons. Upon acquiring the terminal temporal identity, NBs switch to become glial precursor cells (*Zhu et al., 2022*).

At the end of neurogenesis, NBs of different neural lineages are eliminated through distinct mechanisms. For instance, type I NBs in the VNC and the CB exit the cell cycle and terminally differentiate at around 24 hr after pupal formation (APF), whereas type I abdominal NBs are eliminated via apoptosis

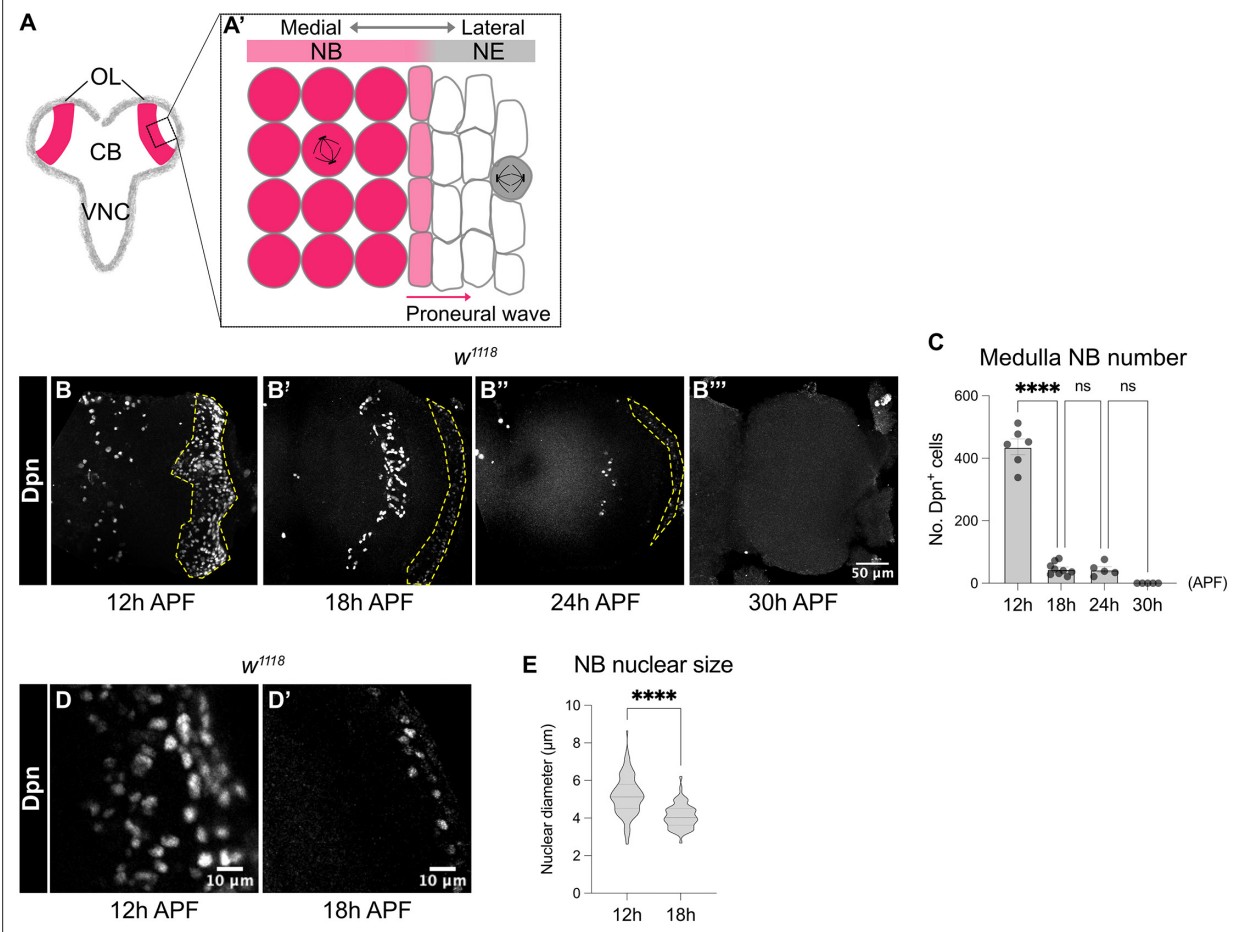

**Figure 1.** Medulla NBs are terminated during early pupal development. (**A-A'**) Schematic depicting the larval CNS that is consisted of three main regions: the central brain (CB), the ventral nerve cord (VNC) and the optic lobe (OL). Box area indicates inset shown in **A'**. (**A'**) The superficial section of the outer proliferation centre (OPC) of the OL. Here, NE cells first symmetrically divide to self-renew. At mid-larval development, a proneural wave is initiated and travels from the medial to lateral of the NE. At the front of the proneural wave, NE cells differentiate into NBs and switch to asymmetric cell division for neurogenesis. (**B-B'''**) Representative maximum projections of the wildtype ($w^{1118}$) OL. The number of medulla NBs marked by Dpn (dashed lines), gradually reduces from 12 hr to 30 hr APF. Medulla NBs completely disappear at 30 hr APF. (**C**) Quantification of the number of medulla NBs at 12 hr, 18 hr, 24 hr, and 30 hr APF in the wildtype OLs. One-way ANOVA and Holm-Šídák's multiple comparisons tests: ****$p<0.0001$. 12 hr: n=6, m=436 ± 25.2. 18 hr: n=9, m=45.2 ± 6.62. 24 hr: n=5, m=43.6 ± 9.26. 30 hr: n=5, m=0.00 ± 0.00. (**D-D'**) Representative single confocal sections of the wildtype OL. The nuclear size of medulla NBs marked by Dpn decreases from 12 hr to 18 hr APF. (**E**) Quantification of the nuclear diameters of medulla NBs at 12 hr and 18 hr APF. Mann-Whitney test: ****$p<0.0001$. 12 hr: n=109 cells, m=5.18 ± 0.099 μm. 18 hr: n=61 cells, m=4.10 ± 0.082 μm.

The online version of this article includes the following figure supplement(s) for figure 1:

**Figure supplement 1.** The termination of medulla NBs does not require cell growth, the Mediator complex and OxPhos.

**Figure supplement 2.** The termination of medulla NBs does not require ecdysone signalling.

prior to pupation (*Bello et al., 2003*; *Ito and Hotta, 1992*; *Maurange et al., 2008*). Mushroom body NBs are the last NBs to undergo termination, and they do so through apoptosis or autophagy prior to adult eclosion (*Pahl et al., 2019*; *Siegrist et al., 2010*). So far, the timing and mode by which medulla NBs terminate are not known; furthermore, it is unclear whether the temporal series, which schedules NB cessation in other NB lineages (*Maurange et al., 2008*; *Syed et al., 2017*) is also involved in the termination of medulla NBs.

Here, we find that medulla NBs terminate between 12 and 18 hr APF. Unlike other NB lineages, the timing of medulla NB cessation is mainly regulated by the depletion of the NE pool. As such, altered NE-NB transition is sufficient to change the timing of NB termination. Additionally, we demonstrate that medulla NBs terminate via a combination of apoptosis, size-symmetric terminal differentiation via Pros and a switch to gliogenesis via Gcm. Pros-mediated differentiation and Gcm-mediated gliogenic

switch occur independently of each other. Notably, while medulla NBs undergo temporal transitions, the temporal series is largely not required for NB termination.

## Results

### Medulla NBs terminate during early pupal development

To characterise the timing at which medulla NBs are eliminated during pupal development, we first monitored the number of NBs in the OPC of $w^{1118}$ animals using the pan-NB marker Deadpan (Dpn). We found that the number of NBs was significantly reduced between 12 and 18 hr APF and by 30 hr APF, all the NBs in the OPC were eliminated (*Figure 1B–C*).

In the larval CB and the VNC, it was previously shown that type I NBs regrow to their original size after each division empowered by aerobic glycolysis which in turn provides the NBs with energy and metabolites for rapid cell growth and proliferation (*Homem et al., 2014*; *Maurange et al., 2008*). However, upon larval-pupal transition, glycolysis is downregulated, resulting in their terminal differentiation (*Homem et al., 2014*). To assess if a reduction in cell size also precedes NB elimination in the medulla, we measured the nuclear size of NBs marked by Dpn between 12 and 18 hr APF when the majority of medulla NBs disappear. We found that there was a significant reduction in the nuclear sizes of the medulla NBs between 12 and 18 hr APF (*Figure 1D–E*), suggesting that decreased cell growth likely precedes the cessation of medulla NB. Nonetheless, increasing cellular growth through the overexpression of a constitutive active form of the PI3K catalytic subunit Dp110 (*Dp110^{CAAX}*; *Leevers et al., 1996*), or the activation of the cellular growth regulator Myc (*Rust et al., 2018*) in clones did not prolong NB persistence in the OLs at 24 hr APF (*Figure 1—figure supplement 1A, B, E, F*). Therefore, although a decrease in cell growth coincides with NB elimination in the pupal medulla, it is not necessary for medulla NB termination.

Previous literature showed that NB termination is dependent on core components of the Mediator complex, which promotes key enzymes to increase oxidative phosphorylation (OxPhos) to slow down NB regrowth (*van den Ameele and Brand, 2019*; *Homem et al., 2014*). We found a significant increase in the number of NBs at 24 hr APF in flip-out clones where we expressed a RNAi against *med12* (*Yang et al., 2017*) which encodes for a component of the Mediator complex (*Figure 1—figure supplement 1A, C, E*). As clonal induction is random and we could not determine whether Med12 is acting in the NE cells or the NBs, we next used a NB-specific driver *eyR16F10-GAL4* (expressed in approximately 50% of medulla NBs, *Figure 1—figure supplement 1G–K*) to drive the expression of *med12 RNAi*. This manipulation did not promote the persistence of medulla NBs at 24 hr APF (*Figure 1—figure supplement 1L–N*), suggesting that *med12* is likely important in the NE but not the NBs. Next, we knocked down *ND75*, a component of the Complex I of the electron transport chain by RNAi (*Granat et al., 2024*) to test the requirement of OxPhos for medulla NB termination. Our data showed that no persistent NBs were found in *ND75 RNAi* clones at 24 hr APF (*Figure 1—figure supplement 1A, D, E*). As such, we conclude that neither the Mediator complex nor OxPhos is required for medulla NB termination during pupal development.

### Ecdysone signalling is not necessary for medulla NB termination during pupal development

In most regions of the CNS, NB termination is scheduled by a peak of ecdysone between larval and pupal development that controls cell growth and proliferation rate of the NBs (*Homem et al., 2014*). Immunostaining of all EcR isoforms showed that EcR was expressed at very low levels in the medulla NBs reported by Dpn::GFP (*Morin et al., 2001*) at 12 hr APF (*Figure 1—figure supplement 2A*). This suggests that ecdysone signalling is likely not active in the pupal medulla NBs. To test if ecdysone signalling is functionally involved in medulla NB termination, we knocked down ecdysone signalling using *eyR16F10-GAL4*. When EcR was knocked down by RNAi (*Schubiger et al., 2005*), we could not detect any persistent NBs at 24 hr APF (*Figure 1—figure supplement 2B, C, E*). A similar phenotype was observed when we utilised a dominant negative form of EcR (*EcR^{DN}*) (*Figure 1—figure supplement 2B, D, E*). Together, these data suggest that ecdysone signalling in the medulla NBs is not required for their termination during pupal development.

## The depletion of the NE determines the timing of medulla NB cessation

Medulla NBs are continuously generated from NE cells from mid-late larval development by the proneural wave (*Egger et al., 2007*; *Yasugi et al., 2008*). To assess if NE depletion during pupal stages is linked to the elimination of medulla NBs, we first performed clonal analysis using the Mosaic Analysis with a Repressible Cell Marker (MARCM) system (*Lee and Luo, 1999*) whereby clones were induced at 24 hr ALH, prior to the NE-NB transition, and then analysed between 12 and 30 hr APF. At 12–18 hr APF, in the OL, clones consisted of both NE (marked by apical PatJ) and NB populations (marked by Dpn) (*Figure 2A–B*). At 24 hr APF, clones consisted of only NE cells (*Figure 2C*) and by 30 hr APF, neither NE cells nor NBs persisted in the clones (*Figure 2D*). Quantifications of the total volumes of NE cells and NBs in the OPC showed that the reduction of NB number preceded the depletion of the NE (*Figure 2E*). Given that the OPC NE also produces lamina precursor cells beside medulla NBs (*Apitz and Salecker, 2015*), it is plausible that a portion of the NE may be preserved for lamina neurogenesis.

To test if the depletion of the NE can change the timing of NB termination, we manipulated Notch signalling, a negative regulator of the NE-NB transition (*Egger et al., 2010*; *Wang et al., 2011*; *Yasugi et al., 2010*), where Notch signalling promotes NE self-renewal and prevents their differentiation into NBs (*Egger et al., 2010*; *Wang et al., 2011*; *Yasugi et al., 2010*). Control and *notch* RNAi (*Xu et al., 2017*) clones were induced in the medulla NE at 24 hr ALH and examined at 72–120 hr ALH. At 72 hr ALH, in the deep section, control clones consisted of only NE cells (*Figure 2—figure supplement 1A*). In contrast, *notch RNAi* clones consisted of few NE cells and ectopic NBs in the deep section of the medulla (*Figure 2—figure supplement 1B*). This indicates the ectopic formation and delamination of NBs from the NE (*Egger et al., 2010*; *Wang et al., 2011*; *Yasugi et al., 2010*). At 120 hr ALH, on the superficial surface of the OL, control clones spanned the PatJ$^+$NE and Dpn$^+$NB regions, demarcated by a sharp NE-NB boundary (*Figure 2F*). Whereas *notch* RNAi clones consisted of mostly NBs with a laterally shifted NE-NB boundary (*Figure 2H*). These are consistent with previous reports suggesting that the downregulation of Notch signalling induces ectopic NE-NB transition and the delamination of NBs into the deeper layers of the medulla (*Egger et al., 2010*; *Wang et al., 2011*; *Yasugi et al., 2010*). At 12 hr APF, we found that *notch* RNAi clones were smaller in size and consisted of very few NBs compared to control (*Figure 2G, I, J*). Thus, it is likely that a precocious NE-NB transition resulted in a premature depletion of the NE, leading to the early termination of medulla NBs. To test this further, we overexpressed the proneural gene *lethal of scute* (*l'sc*), which has been shown to promote NE-NB transition during larval stages (*Yasugi et al., 2008*; *Figure 2K*). Similar to *notch* RNAi clones, *l'sc* overexpression caused a significant reduction in the ratio of NBs in the clones compared to the control at 12 hr APF (*Figure 2L–M*), suggesting a premature elimination of medulla NBs.

Conversely, we found that the overexpression clones of Notch signalling via $N^{ACT}$ caused a medially shifted NE-NB boundary at 120 hr ALH, that was indicative of a delayed NE-NB transition (*Figure 2N*). Furthermore, $N^{ACT}$ clones consisted of only PatJ$^+$NE cells and no Dpn$^+$NBs (*Figure 2N*). At around 72 hr APF, a timepoint where control medulla NBs have long disappeared, we found that $N^{ACT}$ clones appeared overgrown with many persistent ectopic NBs (*Figure 2O*, n=8/8), indicating tumour formation. Overall, our results suggest that early depletion of the NE results in the early cessation of medulla NBs, whereas delayed formation of the NBs from the NE results in the late termination of medulla NBs. As such, the timing of the NE-NB transition likely determines when medulla NBs terminate.

## Medulla NBs cease divisions via apoptosis-mediated cell death

Next, we investigated how medulla NBs undergo termination. First, we examined mechanisms responsible for the termination of other NB lineages. Mushroom Body NBs are the last NBs to undergo termination during late pupal development (*Ito and Hotta, 1992*) via a combination of apoptosis and autophagy (*Siegrist et al., 2010*). To test if medulla NBs terminate via similar mechanisms, we first assayed for apoptosis using the apoptotic marker Caspase-1 (Dcp-1) in the medulla NBs marked by Dpn::GFP at 12 hr and 16 hr APF, the timepoints between which we observed a significant reduction in the number of medulla NBs (*Figure 1C*). We found many NBs expressed Dcp-1 at these two timepoints (*Figure 3A–J*), suggesting that medulla NBs undergo apoptosis during early pupal stages.

In the OL, medulla NBs sequentially express an array of temporal transcription factors as they age to promote cellular diversity of their progeny (*Konstantinides et al., 2022*; *Li et al., 2013*; *Zhu et al., 2022*). The simplified temporal series includes Homothorax (Hth), Eyeless (Ey), Sloppy-paired (Slp),

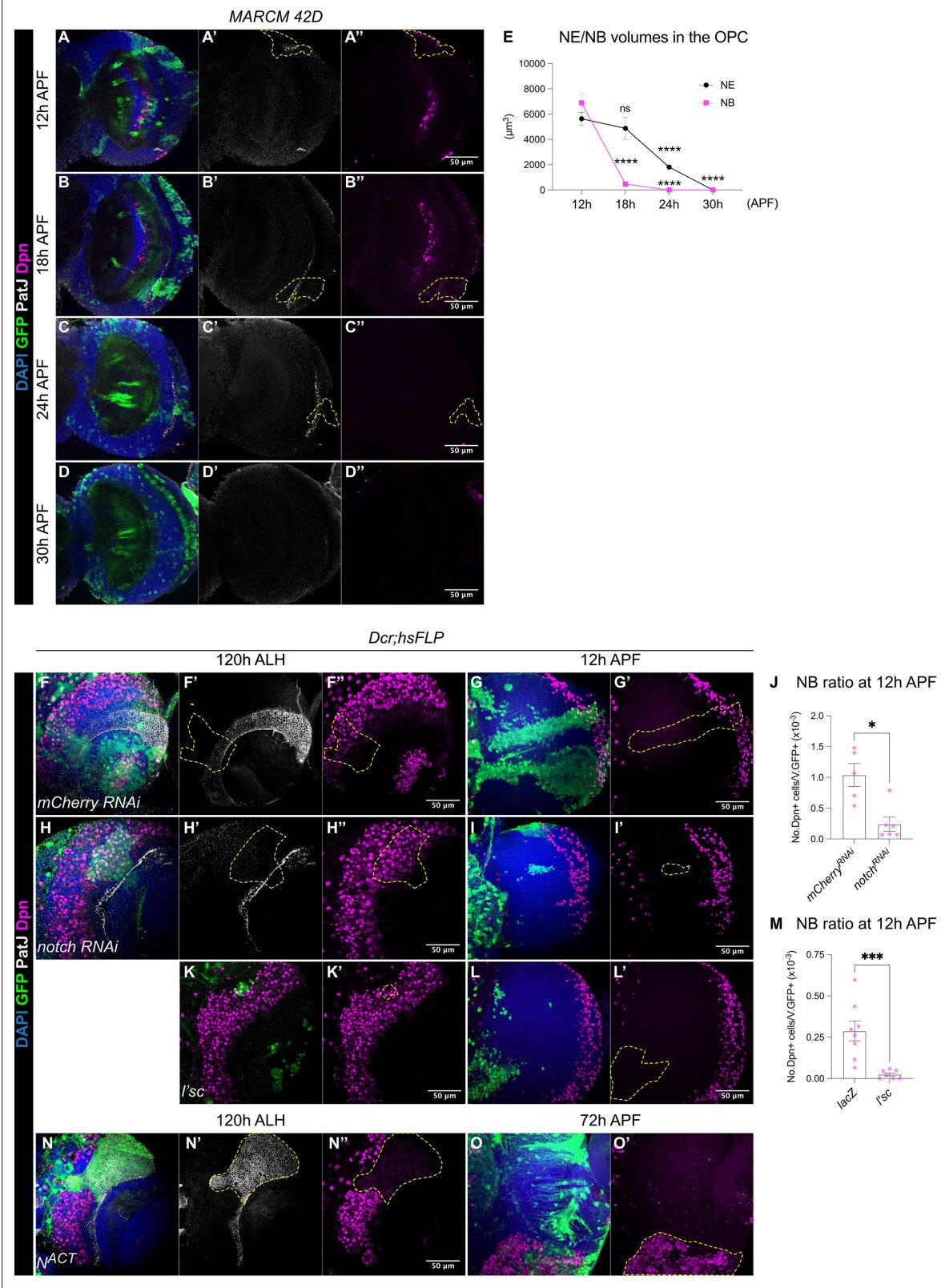

**Figure 2.** The termination of medulla NBs is scheduled by the timing of the larval NE-NB transition. (**A-D**) Representative single confocal sections of the OL, where MARCM clones show that between 12 hr and 30 hr APF the gradual depletion of the NE (marked by PatJ, grey) is accompanied by the elimination of NBs (marked by Dpn, magenta) in the medulla. Dashed lines indicate representative clones (marked by GFP, green), DAPI (blue). Note that at 24 hr and 30 hr APF, neurons migrate and become dispersed across the OL, therefore, the precise annotation of the clones is not

*Figure 2 continued on next page*

*Figure 2 continued*

possible. (**E**) Quantification of NE and NB volumes at 12 hr, 18 hr, 24 hr, and 30 hr APF. One-way ANOVA and Holm-Šídák's multiple comparisons tests: ****$p<0.0001$. (NE) 12 hr: m=5635 ± 502. 18 hr: 4877±884. 24 hr: 1814±190. 30 hr: 0.00±0.00. (NB) 12 hr: 6892±769. 18 hr: 472±80.1. 24 hr: 0.00±0. 30 hr: 0.00±0.00. (**F, H**) Representative single confocal sections of the OLs at 120 hr ALH in which *hsFLP* clones (dashed lines) express (**F-F"**) *UAS-mCherry RNAi* and (**H-H"**) *UAS-notch RNAi*. DAPI (blue), GFP (green), PatJ (grey), Dpn (magenta). (**G, I**) Representative maximum projections of the OL at 12 hr APF, in which *hsFLP* clones (dashed lines) express (**G-G'**) *UAS-mCherry RNAi* and (**I-I'**) *UAS-notch RNAi*. (**J**) Quantification of the NB ratio in the OL at 12 hr APF in which *hsFLP* clones were induced with *UAS-mCherry RNAi* and *UAS-notch RNAi*. Mann-Whitney test: *$p=0.0173$. *mCherry RNAi*: n=5, m=1.04 ± 0.185. *notch RNAi*: n=6, m=0.241 ± 0.113. (**K-K'**) Representative single confocal sections of the OLs at 120 hr ALH in which *hsFLP* clones (dashed line) express *UAS-l'sc*. (**L-L'**) Representative maximum projections of the OL (dashed lines) at 12 hr APF in which *hsFLP* clones express *UAS-l'sc*. (**M**) Quantification of the NB ratio in *UAS-lacZ* and *UAS-l'sc* clones in the OL at 12 hr APF. Mann-Whitney test: ***$p=0.0003$. *lacZ*: n=8, m=0.288 ± 0.061. *l'sc*: n=7, m=0.025 ± 0.010. (**N-N'**) Representative single confocal sections of the OLs at 120 hr ALH in which *hsFLP* clones (dashed line) express *UAS-N^{ACT}*. (**O-O'**) Representative maximum projections of the OL at 72 hr APF, in which *hsFLP* clones (dashed line) express *UAS-N^{ACT}*. Note that *N^{ACT}* clones do not generate NBs until late pupal development.

The online version of this article includes the following figure supplement(s) for figure 2:

**Figure supplement 1.** The downregulation of Notch signalling in the NE causes precocious formation of medulla NBs.

Dichaete (D) and Tailless (Tll) (*Figure 3—figure supplement 1A*). To assess if apoptosis of the medulla NBs is associated with the temporal series, we looked at the expression of three temporal factors Ey, Slp, and Tll in apoptotic medulla NBs at 12 hr and 16 hr APF. At 12 hr APF, we found that half of the apoptotic medulla NBs expressed Ey, whereas very few NBs expressed Slp or Tll (*Figure 3A–D and K*). By 16 hr APF, very few apoptotic medulla NBs expressed any of these temporal factors (*Figure 3E–K*). As such, it is likely that many medulla NBs that undergo apoptosis exhibit a mid-temporal identity (during the Ey⁺ window).

We next tested if apoptosis is required for the termination of medulla NBs by inhibiting apoptosis via the overexpression of the baculovirus anti-apoptotic gene *p35* in clones as well as in medulla NB lineages (*eyR16F10-GAL4*). We observed a small number of persistent NBs at 24 hr APF upon the inhibition of apoptosis (*Figure 3L–P*, *Figure 3—figure supplement 1B–C, G*). However, no persistent NBs were found at 48 hr APF (*Figure 3—figure supplement 1F*, n=5/5), suggesting that additional factors may be involved in terminating medulla NBs.

We then asked if autophagy is required to eliminate medulla NBs similarly to mushroom body NBs (*Siegrist et al., 2010*). To do so, we inhibited autophagy via the expression of a RNAi against *Atg1* (*Bierlein et al., 2023*) in the medulla NBs with *eyR16F10-GAL4*. At 24 hr APF, no persistent NBs were recovered in the OL upon *Atg1 RNAi* expression (*Figure 3—figure supplement 1D, G*). Furthermore, concomitant inhibitions of apoptosis and autophagy did not further enhance NB persistence compared to the inhibition of apoptosis alone (*Figure 3—figure supplement 1E, G*). Together, our data suggest that apoptosis, and not autophagy, is required to terminate a subset of medulla NBs during early pupal development.

## Medulla NBs undergo Pros-mediated termination

In type I NBs of the VNC, the nuclear localisation of a differentiation factor called Prospero (Pros) precedes NB termination via size-symmetric cell division (*Choksi et al., 2006*; *Li et al., 2013*; *Maurange et al., 2008*). To assess if medulla NBs also undergo Pros-mediated terminal differentiation, we first examined Pros expression together with the pan-NB reporter Dpn::GFP at 12 hr APF. We found that many Dpn⁺ NBs expressed nuclear Pros (*Figure 4A*). Additionally, we examined the expressions of Dpn::GFP with Miranda (Mira) that marks mature NBs and asymmetric cell division at 12 hr APF. Here, we occasionally observed some small Dpn⁺ Mira⁻ cells, that formed doublets at the most medial part of the OPC (*Figure 4B*), which we think, are likely to be progenies of NB symmetric divisions. Therefore, it is possible that medulla NBs also undergo size symmetric divisions via Pros nuclear localisation like those of other lineages.

To test if Pros is sufficient to induce the termination of medulla NBs, we overexpressed *pros* using *eyR16F10-GAL4* and assessed the total number of NBs in the medulla at 120 hr ALH. To avoid animal lethality, we expressed *pros* for 2 days during late L3, using a temperature sensitive *GAL80^{ts}* together with *eyR16F10-GAL4*. Upon *pros* overexpression, the number of medulla NBs was reduced by around 50% compared to the control (*Figure 4C–E*), suggesting that Pros is sufficient to induce premature medulla NB termination.

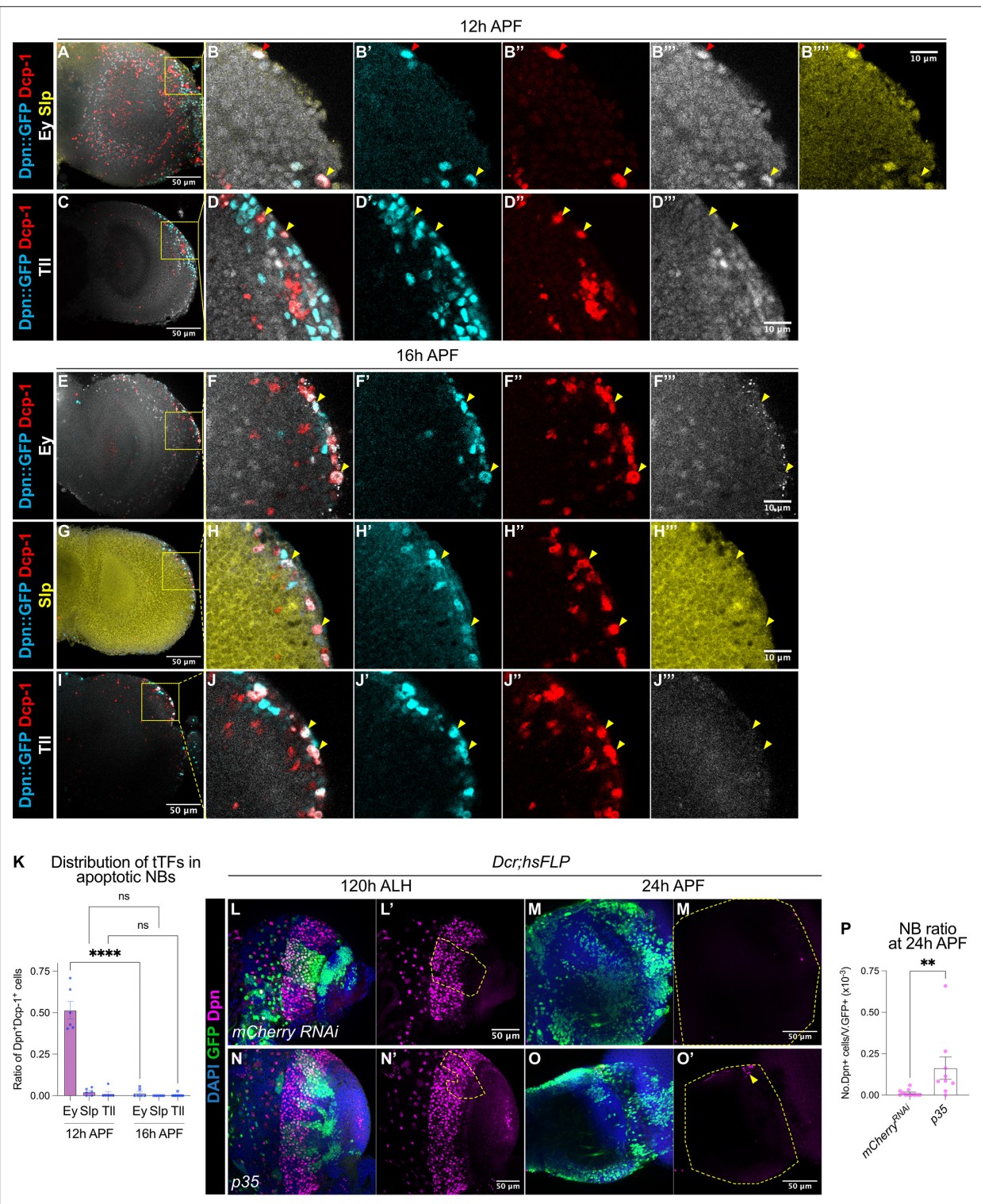

**Figure 3.** A subset of medulla NBs terminate via apoptosis-mediated cell death. (**A–B**) Representative maximum projections of the OL at 12 hr APF. A subset of medulla NBs marked by Dpn::GFP (cyan) express the apoptotic marker Dcp-1 (red). **B-B''''** is the magnified view of the boxed inset in A. Among these, some cells express Ey (grey) (red arrowhead) and a smaller population express Slp (yellow) (yellow arrowhead). (**C–D**) Representative maximum projections of the OL at 12 hr APF. Medulla NBs marked by Dpn::GFP (cyan) expressing Dcp-1 (red) do not express Tll (grey) (yellow arrowheads). **D-D''''** is the magnified view of the boxed inset in C. (**E–J**) Representative maximum projections of the OL at 16 hr APF. Medulla NBs marked by Dpn::GFP (cyan) expressing Dcp-1 (red) express neither Ey, Slp, nor Tll (yellow arrowheads). (**F, H, J**) are the magnified view of the boxed insets in (**E, G, I**) respectively. Ey or Tll (grey), Slp (yellow). (**K**) Quantifications of the ratio of apoptotic NBs in the medulla expressing different temporal

*Figure 3 continued on next page*

*Figure 3 continued*

Transcription Factors (tTFs) at 12 hr and 16 hr APF. Two-way ANOVA test with Šídák's multiple comparisons: ****p<0.0001. 12 hr: Ey: n=6, m=0.515 ± 0.053. Slp: n=6, m=0.023 ± 0.008. Tll: n=6, m=0.012 ± 0.012. 16 hr: Ey: n=6, m=0.015 ± 0.010. Slp: n=6, m=0.00 ± 0.00. Tll: n=7, m=0.004 ± 0.004. (**L, N**) Representative maximum projections of the OLs at 120 hr ALH with *hsFLP* clones (dashed lines) expressing (**L-L'**) *UAS-mCherry RNAi* and (**N-N'**) *UAS-p35*. DAPI (blue), GFP (green), Dpn (magenta). (**M, O**) Representative maximum projections of the OLs at 24 hr APF with *hsFLP* clones expressing (**M-M'**) *UAS-mCherry RNAi* and (**O-O'**) *UAS-p35*. The OLs are outlined by dashed lines. DAPI (blue), GFP (green), Dpn (magenta). The same *mCherry RNAi* representative image is used in *Figure 5L*. (**P**) Quantification of the NB ratio in *hsFLP* clones expressing *UAS-mCherry RNAi* and *UAS-p35* in the OLs at 24 hr APF. Mann-Whitney test: **p=0.002. *mCherry RNAi*: n=11, m=0.012 ± 0.006. *p35*: n=9, m=0.163 ± 0.067.

The online version of this article includes the following figure supplement(s) for figure 3:

**Figure supplement 1.** Apoptosis but not autophagy is necessary for the termination of medulla NBs.

To test if Pros is required for the termination of medulla NBs, we induced flip-out *pros* RNAi clones (*Shaw et al., 2018*). This manipulation caused the formation of ectopic NBs in the deep sections of the medulla where neurons reside during larval development (*Figure 4F*). By 24 hr APF, many persistent NBs were found in *pros RNAi* clones compared to the control (*Figures 3M, 4G and I*). However, by 48 hr APF, we recovered very few persistent NBs in *pros RNAi* clones (*Figure 4H*). Interestingly, by pupal stages, the *pros RNAi* clones were small and dispersed, in addition, most of the cells within the clone were devoid of Dpn expression (*Figure 4G–H*), indicating that the NBs may have undergone differentiation. Altogether, our data suggest that Pros is sufficient and necessary to promote medulla NB termination (*Figure 4J*). However, it is likely that medulla NBs can also undergo termination via Pros-independent mechanisms.

## Gcm-induced gliogenic switch can affect the timing of NB termination

Prior work suggest that at the end of the temporal series, larval medulla NBs express the glial cell determinant Gcm following the Tll[+] window to induce NB differentiation into glial cell precursors (*Konstantinides et al., 2022*; *Li et al., 2013*; *Zhu et al., 2022*). At 12 hr APF, we observed some superficially located Dpn[+] NBs which only expressed the differentiation marker Pros::GFP (*Figure 5A*, red arrowheads). In addition, we observed some cells which were double positive for Pros::GFP and *gcm*-lacZ (*Figure 5A*, yellow arrowheads), suggesting that these NBs could be undergoing a NB-to-glioblast cell fate transition. Deeper within the medulla, we also found *gcm*[+] cells which expressed neither Dpn nor Pros::GFP (*Figure 5A*, white arrowhead), that were likely the glial progeny.

Because we observed cells in the OL which had overlapping Pros and *gcm* expression (*Figure 5A*), we next tested whether Pros is required for glial cell fate acquisition by the medulla NBs. To do so, we generated *pros RNAi* or *pros*[17] (*Doe et al., 1991*) clones, and assessed for glial cells marked by Repo at 120 hr ALH. We found that the ratio of glia recovered was not significantly affected by the down-regulation or the loss-of-function of Pros (*Figure 5B–D* and *Figure 5—figure supplement 1A–B*). Together, these results suggest that *pros* is not necessary for the neurogenic-to-gliogenic switch in the medulla NBs.

Next, we tested if manipulating the gliogenic switch can affect the timing of medulla NB termination. First, we induced *gcm* overexpression clones using a reagent previously validated in *Zhu et al., 2022*. Indeed, this manipulation was able to induce the formation of ectopic glial cells during larval neurogenesis (*Figure 5E and G*). At 12 hr APF, *UAS-gcm* clones showed a significantly lower ratio of NBs compared to the control (*Figure 5F, H, I*), suggesting that the induction of a gliogenic switch can cause precocious medulla NB termination. Then, we tested if inhibition of gliogenesis can cause prolonged NB persistence in the medulla. We expressed *gcm RNAi* (*Zhu et al., 2022*) in clones which suppressed the expression of the glial cell marker Repo during larval stages (*Figure 5J and M*). By 12 hr APF, *gcm RNAi* clones showed a significantly higher ratio of NBs compared to the control (*Figure 5K, N and P*), suggesting that the inhibition of glial cell fate can promote medulla NB persistence. However, by 24 hr APF, no persistent NBs were observed in *gcm RNAi* clones (*Figure 5L and O*, n=5/5). As we have shown that medulla NBs can also terminate via Pros-mediated size symmetric divisions, we wondered whether this mechanism remains active upon the inhibition of gliogenesis. Indeed, Pros remained unaltered in *gcm RNAi* clones (*Figure 5—figure supplement 1C–D*), suggesting that Gcm-mediated gliogenesis and Pros-mediated differentiation occur in parallel to promote the termination of medulla NBs.

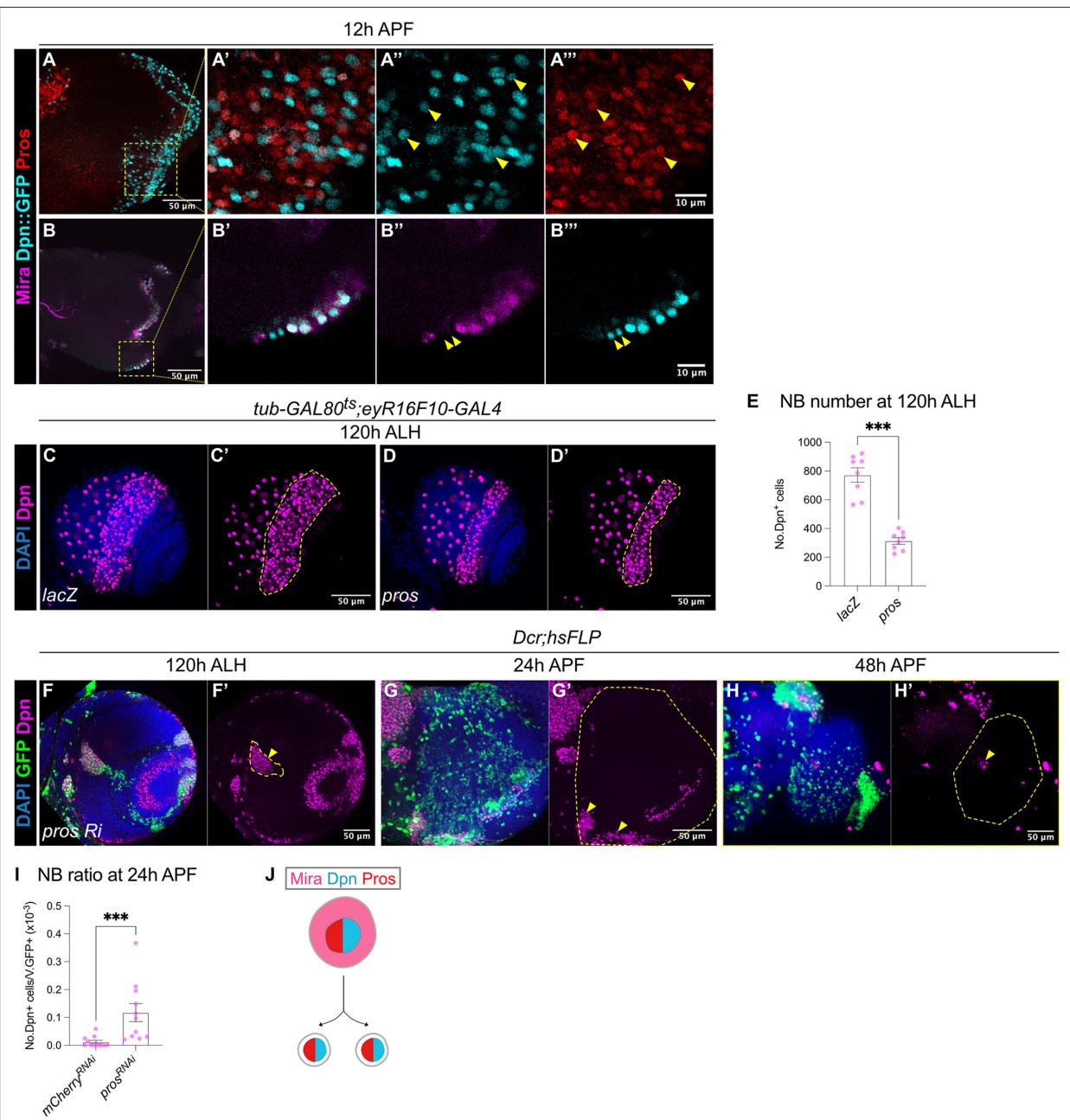

**Figure 4.** The termination of medulla NBs requires Pros-mediated size symmetric divisions. (**A-A'''**) Representative maximum projection of the OL at 12 hr APF. Some NBs marked by Dpn::GFP (cyan) in the medulla co-express Pros (red) (arrowheads). **A'-A'''** are magnified images of boxed inset in A. (**B-B'''**) Representative single confocal section of the OL at 12 hr APF. On the superficial layer where medulla NBs reside, we have occasionally observed a pair of similarly sized cells at the most medial position (arrowheads), that express Dpn::GFP (cyan) but not Mira (magenta). **B'-B'''** are magnified images of boxed inset in B. (**C–D**) Representative maximum projections of the OLs at 120 hr ALH in which *tub-GAL80^ts;eyR16F10-GAL4* drives (**C-C'**) *UAS-lacZ* and (**D-D'**) *UAS-pros.* Dashed lines outline medulla NBs. DAPI (blue), Dpn (magenta). (**E**) Quantifications of medulla NB numbers at 120 hr ALH where *tub-GAL80^ts;eyR16F10-GAL4* drives *UAS-lacZ* and *UAS-pros.* Mann-Whitney test: ***p=0.0002. *lacZ*: n=8, m=772 ± 50.4. *pros*: n=8, m=313 ± 22.7. (**F**) Representative single confocal section of the OL showing the deep section of the medulla with *hsFLP* clones (dashed lines) expressing *UAS-pros RNAi* at 120 hr ALH. Arrowhead indicates ectopic NBs. DAPI (blue), GFP (green), Dpn (magenta). (**G–H**) Representative maximum projects of an OL (dashed line) at (**G-G'**) 24 hr APF and (**H-H'**) 48 hr APF, with *hsFLP* clones expressing *UAS-pros RNAi.* Arrowheads indicate persistent NBs. DAPI (blue), GFP (green), Dpn (magenta). (**I**) Quantification of the NB ratio in *hsFLP* clones in the OL at 24 hr APF, expressing *UAS-mCherry RNAi* and *UAS-pros RNAi.* Mann-Whitney test: ***p=0.0003. *mCherry RNAi*: n=11, m=0.012 ± 0.006. *pros RNAi*: n=11, m=0.117 ± 0.033. (**J**) Schematic depicting a possible model of differentiation: Mira⁺Dpn⁺Pros⁺ NB gives rise to two Dpn⁺Pros⁺ progeny.

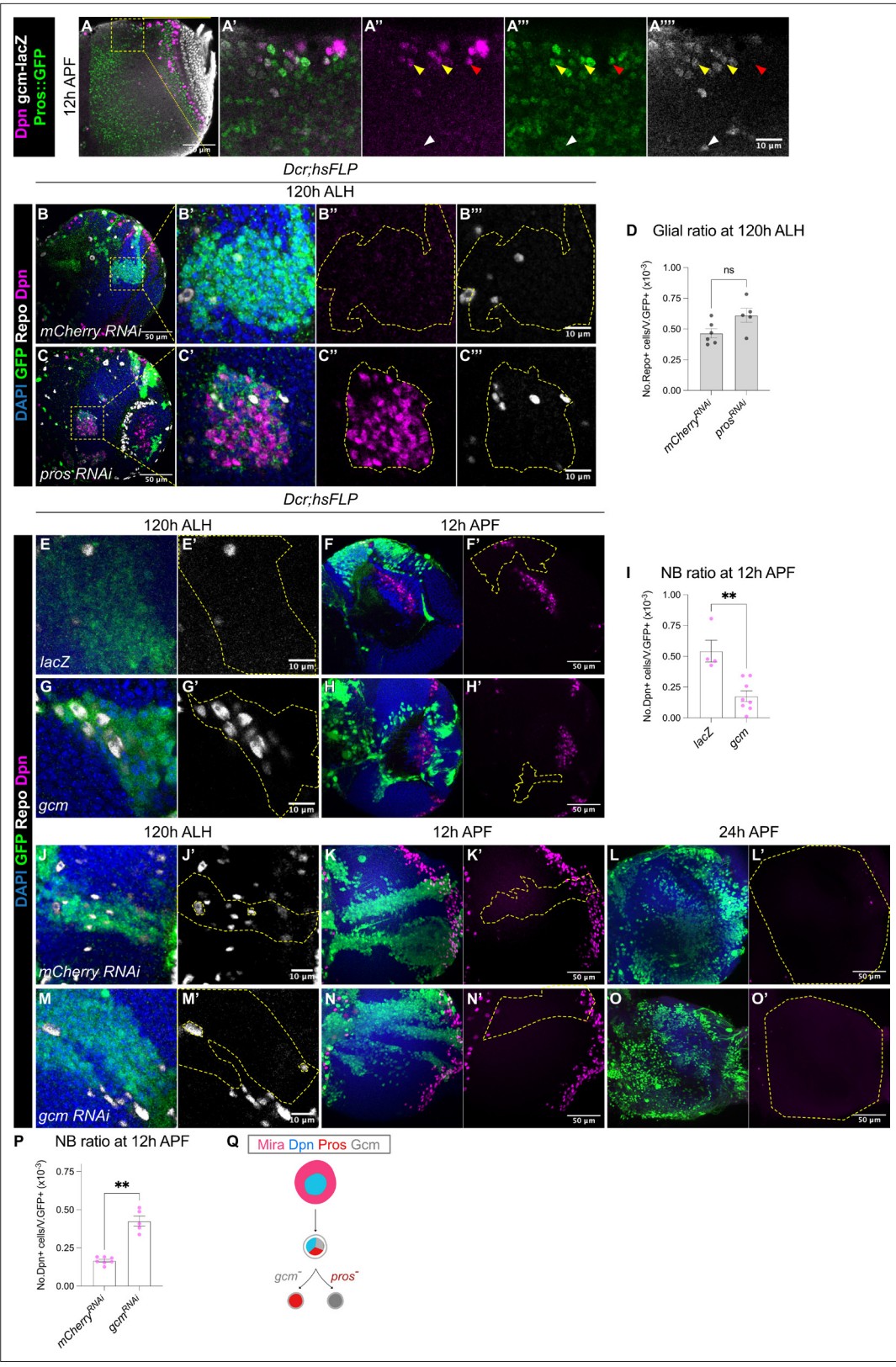

**Figure 5.** Gcm is sufficient to induce gliogenesis in the medulla at the expense of NBs. (**A-A''''**) Representative single confocal section of the OL at 12 hr APF. In the superficial layer, some of the most medial NBs are Dpn⁺Pros⁺*gcm*⁺ (yellow arrowheads), while some NBs are Dpn⁺Pros⁺*gcm*⁻ (red arrowhead). In the deep layers, some cells are Dpn⁻Pros⁻*gcm*⁺ (white arrowheads). (**B–C**) Representative single confocal sections of the OLs at 120 hr

*Figure 5 continued on next page*

*Figure 5 continued*

ALH, with *hsFLP* clones (dashed lines) induced with (**B-B'''**) *UAS-mCherry RNAi* and (**C-C'''**) *UAS-pros RNAi*. DAPI (blue), GFP (green), Dpn (magenta), Repo (grey). (**D**) Quantification of glia cell ratio in *hsFLP* clones in the medulla at 120 hr ALH that express *UAS-mCherry RNAi* and *UAS-pros RNAi*. Mann-Whitney test: (ns) p=0.052. *mCherry RNAi*: n=6, m=0.466 ± 0.037. *pros RNAi*: n=5, m=0.612 ± 0.057. (**E, G**) Representative single confocal sections of *hsFLP* clones (dashed lines) at 120 hr ALH, expressing (**E-E'**) *UAS-lacZ*, and (**G-G'**) *UAS-gcm*. DAPI (green), GFP (green), Repo (grey). (**F, H**) Representative images of the OLs (dashed lines) at 12 hr APF, in which *hsFLP* clones express (**F-F'**) *UAS-lacZ*, and (**H-H'**) *UAS-gcm*. DAPI (blue), GFP (green), Dpn (magenta). (**I**) Quantification of the NB ratio in *hsFLP* clones in the OLs at 12 hr APF, that express *UAS-lacZ* and *UAS-gcm*. Mann-Whitney test: **p=0.004. *lacZ*: n=4, m=0.543 ± 0.088. *gcm*: n=8, m=0.176 ± 0.044. (**J, M**) Representative single confocal sections of *hsFLP* clones (dashed lines) at 120 hr ALH, expressing (**J-J'**) *UAS-mCherry RNAi*, and (**M-M'**) *UAS-gcm RNAi*. DAPI (green), GFP (green), Repo (grey). (**K, L, N, O**) Representative maximum projections of the OLs (dashed lines) at (**K, N**) 12 hr APF and (**L, O**) 24 h APF, in which *hsFLP* clones express (**K-K'**, **L-L'**) *UAS-mCherry RNAi*, and (**N-N'**, **O-O'**) *UAS-gcm RNAi*. DAPI (blue), GFP (green), Dpn (magenta). (**P**) Quantification of the NB ratio in *hsFLP* clones in the OLs at 12 hr APF, that express *UAS-mCherry RNAi* and *UAS-gcm RNAi*. Mann-Whitney test: **p=0.003. *mCherry RNAi*: n=7, m=0.167 ± 0.009. *gcm RNAi*: n=5, m=0.426 ± 0.033. (**Q**) Schematic depicting a Mira$^+$Dpn$^+$ medulla NB expressing nuclear Pros and the glial cell fate determinant Gcm to either produce Gcm$^+$ glial cells, or Pros$^+$ neurons. Pros expression is not affected by the absence of Gcm, and Gcm expression is also not affected by the absence of Pros.

The online version of this article includes the following figure supplement(s) for figure 5:

**Figure supplement 1.** Pros-mediated size symmetric division and Gcm-mediated gliogenesis are independent mechanisms that promote medulla NB termination.

**Figure supplement 2.** Tll is not required to schedule the gliogenic switch of the medulla NBs.

Collectively, our data suggest that *gcm* overexpression can trigger a premature neurogenic-to-gliogenic switch and earlier termination of medulla NBs (***Figure 5Q***). However, *gcm* downregulation is not sufficient to prolong medulla NB persistence likely due to redundant termination mechanisms such as Pros-mediated size-symmetric divisions.

## Temporal progression plays a minor role in the control of medulla NB termination

It was previously shown that nuclear Pros localisation in the medulla NBs occurred during the Tll$^+$ temporal window (***Li et al., 2013***). To test if members of the temporal series can alter NB persistence and prevent the neurogenic-to-gliogenic switch, we first examined the role of Tll, which defines the temporal window where terminal differentiation as well as the gliogenic switch occur (***Li et al., 2013***; ***Zhu et al., 2022***). First, we induced *tll* overexpression clones and found that it caused the formation of ectopic NBs persistent in the OL at 24 hr APF (***Figure 5—figure supplement 2B***, n=4/4). *tll* overexpression has been previously shown to convert type I to type II NB cell fate that is capable of tumour formation in type I NBs of the CB (***Hakes and Brand, 2020***). Indeed, *tll* overexpressing NBs in the larval and pupal OLs were mostly Dpn$^+$Ase$^-$ which is characteristic of type II NBs (***Figure 5—figure supplement 2A–B***). Therefore, it is possible that *tll* overexpression caused type I to type II conversion in the NBs that accounts for their failure to undergo termination in a timely manner.

Consistent with previous reports that gliogenesis occurs in the NBs of the Tll$^+$ window but Tll is not necessary for this process (***Li et al., 2013***; ***Zhu et al., 2022***), we found that at 12 hr APF, multiple medulla NBs co-expressed Tll::EGFP and *gcm*-lacZ (***Figure 5—figure supplement 2C***). However, the inhibition of *tll* via a validated RNAi (***Hakes and Brand, 2020***) in clones neither altered the number of glial cells in the larval medulla (***Figure 5—figure supplement 2D, E, H***), nor the timing of medulla NB termination at 12 hr APF (***Figure 5—figure supplement 2F, G, I***). Moreover, the knockdown of *tll* did not affect the number of NBs undergoing apoptosis assayed via Dcp-1 expression (***Figure 5—figure supplement 2J–K***), consistent with our data showing that most apoptotic medulla NBs at 12–16 hr APF did not express Tll (***Figure 3K***). Additionally, *tll* knockdown did not alter Pros expression which promotes terminal size-symmetric divisions (***Figure 5—figure supplements 1C and 2L***). Altogether, while Tll is not required for medulla NB termination, ectopic expression of Tll can induce persistent NBs. Furthermore, Tll acts independently of apoptosis and symmetric NB termination mechanisms.

To assess whether other temporal transcription factors are required for medulla NB termination, we overexpressed Hth and D (***Konstantinides et al., 2022***; ***Zhu et al., 2022***) in clones to prevent

**Figure 6.** Working model of the termination of medulla NBs. In the wildtype, medulla NBs terminate during early pupal development via three main mechanisms: (1) apoptosis, (2) Pros-mediated symmetric division, and (3) Gcm-mediated gliogenic switch. It is of note that Pros and Gcm function independently to promote medulla NB termination. Timing of NB termination in the early pupal stages is scheduled by the timing of the NE-NB transition in the larval stages. Upon the expression of $N^{ACT}$, NE cells overgrow resulting in a delayed NE-NB transition and persistent NBs. Conversely, expression of *notch RNAi* leads to a precocious NE-NB transition, leading to a premature NB termination.

temporal progression of the NBs. None of these manipulations significantly influenced the timing of NB termination (*Figure 5—figure supplement 2M*, n=3/11, *Figure 5—figure supplement 2N*, n = 6/11). As we have not exhaustively tested the role of all the temporal transcription factors, we conclude that the temporal series makes a minor contribution to the timing of medulla NB termination.

## Discussion

Robust brain development requires the tight coordination between stem cell maintenance and differentiation. In the *Drosophila* OL, there are two main stem cell pools, the NE cells that divide symmetrically, and the NBs. Medulla NBs are derived from the OPC NE via the proneural wave, which switch to asymmetric division where they self-renew and give rise to GMCs that will produce post-mitotic neurons and glial cells of the medulla cortex (*Egger et al., 2007*; *Li et al., 2013*; *Yasugi et al., 2008*). Our study shows that during early pupal development, medulla NBs terminate via a combination of mechanisms including apoptosis-mediated cell death, Pros-mediated differentiation and a switch from neurogenesis to gliogenesis. While factors such as ecdysone signalling and OxPhos of the NBs are involved in NB termination in other lineages outside of the OL (*Homem et al., 2014*; *Syed et al., 2017*; *van den Ameele and Brand, 2019*; *Yang et al., 2017*), we found that these factors do not play a significant role in the termination of medulla NBs. Instead, the main regulator of the timely NB cessation in the pupal medulla is the timing of the NE-NB conversion during larval stages (*Figure 6*). Interestingly, we showed that while *med12 RNAi* clones exhibited persistent NBs in the OL at 24 hr APF, this is likely due to the role of Med12 in regulating NE-NB transition during larval development (Med12 knockdown in the medulla NBs did not cause persistent NBs; *Figure 1—figure supplement 1*). Additionally, while a specific cascade of temporal identity transcription factors in the medulla NBs has been shown to generate the cellular diversity of medulla neurons and glia (*Konstantinides et al., 2022*; *Li et al., 2013*; *Zhu et al., 2022*), temporal progression in the medulla NBs appears to exert negligible effects on the timing of NB cessation in the pupal stages. Diseases such as microcephaly are mediated by the premature depletion of the NE pool, or through a precocious commitment to neurogenic divisions (*Götz and Huttner, 2005*). Consistent with this, our data present evidence that during development, the regulation of the NE stem cell pool, and its timely conversion into NSCs is of critical importance for the overall size and function of the brain.

### NE-NB transition determines the timely termination of medulla NBs

Our data support a model whereby once the NE pool is exhausted, no new NBs can be generated. NBs transit through their temporal series to terminally differentiate into neurons and glia, which marks the end of medulla neurogenesis. Thus, unlike type I NB lineages in which NB termination is scheduled by ecdysone signalling-mediated temporal progression at the level of the NBs (*Syed et al., 2017*; *Yang et al., 2017*); in the OL, NB termination is pre-determined in the NE. While our work has only tested the role of Notch signalling in the NE for determining medulla NB termination, JAK/STAT signalling was previously demonstrated to control the proneural wave and the associated NE-NB

transition, in part through Notch signalling (*Yasugi et al., 2008*; *Yasugi et al., 2010*). Therefore, we speculate that JAK/STAT signalling will also likely influence the timing of medulla NB termination. In the OL, besides medulla NBs, NE cells also sequentially express an array of temporal factors, such as Chinmo and Broad downstream of ecdysone signalling (*Arain et al., 2022*; *Dillard et al., 2018*; *Lanet et al., 2013*; *Zhou et al., 2019*). Moreover, inhibiting the Chinmo-to-Broad expression switch in the NE was shown to delay the NE-NB transition, resulting in tumours with persistent NBs in the adult OLs (*Dillard et al., 2018*; *Zhou et al., 2019*). Overall, the differences in the regulation of NB termination in the OL medulla versus other lineages may be due to some fundamental differences in the origins of these NBs. In the OL, NBs are derived from the NE, that undergoes rapid growth during larval development (*Egger et al., 2007*; *Lanet et al., 2013*; *Yasugi et al., 2008*). In contrast, NBs from other lineages are produced directly from the neuroectoderm in the embryo without undergoing the transition from the NE (*Harding and White, 2018*). Hence, their termination mechanism acts predominantly in the NBs during pupal development.

## Apoptosis-mediated cell death makes a small contribution to NB termination

Consistent with previous reports, we found that apoptosis occurred in a subset of medulla NBs from 12 hr to 16 hr APF (*Hara et al., 2018*; *Figure 3*). Furthermore, we showed that although apoptosis inhibition caused persistent NBs at 24 hr APF, by 48 hr APF, all NBs were still eliminated in the OL (*Figure 3M, O and P*, *Figure 3—figure supplement 1B–G*). Notably, half of the NBs undergoing apoptosis at 12 hr APF expressed the mid-temporal factor Ey, but not Slp or Tll (*Figure 3A–D and K*). These data suggest an intriguing link between the expression of the mid-temporal factor and the competence of the NBs to undergo apoptosis. It is possible that the early-born medulla NBs that in turn express late temporal factors (Tll or Gcm) during pupal development, will terminate via either Pros-mediated symmetric division or gliogenic switch. Whereas the late-born NBs, that still express mid-temporal transcription factors such as Ey, will terminate via apoptosis. This mechanism could be a fail-safe paradigm to ensure the elimination of all the medulla NBs before adulthood. Whether such decision of the NBs is determined by the intrinsic factors like the temporal series, or extrinsic factors such as signalling cues from other cell types in the niche remain to be probed. Although the timing of apoptosis in the medulla NBs coincides with a small pulse of ecdysone around 10 hr APF (*Riddiford, 1993*), we showed that the inhibition of ecdysone signalling in the medulla NBs did not phenocopy the inhibition of apoptosis (*Figure 1—figure supplement 2B–E*, *Figure 3—figure supplement 1C, G*). Therefore, unlike neurons in the OL which require ecdysone signalling for apoptosis during pupal development (*Hara et al., 2013*), medulla NBs undergo apoptosis independently of ecdysone signalling. Altogether, while some NBs terminate proliferation through apoptosis, it is likely that apoptosis of the NBs makes a minor contribution to the termination of neurogenesis in the medulla.

## Symmetric divisions and a neurogenic-gliogenic switch cause NB cessation

We showed that many medulla NBs displayed nuclear Pros expression during early pupal stages (*Figure 4A*). Consistent with the essential role of Pros in promoting differentiation (*Choksi et al., 2006*; *Maurange et al., 2008*), *pros* knockdown caused the persistence of the NBs in the OL at 24 hr APF (*Figure 4G, I*). Nevertheless, later during the pupal stages, *pros* knockdown was insufficient to maintain medulla NB persistence (*Figure 4H*), suggesting that Pros expression in the medulla NBs is not strictly required for their termination. In addition to Pros-mediated differentiation, another mechanism responsible for the cessation of neurogenesis is the gliogenic switch. We found cells triple positive for the pan-NB marker Dpn, the differentiation marker Pros, and the glial cell determinant Gcm in the early pupal medulla (*Figure 5A*), which likely give rise to glial cells at the end of the temporal series. Interestingly, the inhibition of Pros did not affect gliogenesis in the larval medulla (*Figure 5B–D*, *Figure 5—figure supplement 1A–B*); conversely, the inhibition of Gcm did not influence Pros expression in the pupal OL (*Figure 5—figure supplement 1C–D*). As such, these data suggest that despite Pros and Gcm co-localisation in the terminating NBs, gliogenesis and differentiation appear to act in parallel. Taken together, apoptosis, symmetric cell division and a gliogenic switch appear to be employed as fail-safe mechanisms that help eliminate the NB progenitor pool in

the pupal OL (*Figure 6*). Hence, it would be interesting to explore in the future the cross-regulatory relationship between these termination programs.

## Materials and methods
### Fly stocks and husbandry
The fly strains used were detailed in the following table:

| System | Genotype | Source |
|---|---|---|
| Wildtype | *w1118* | |
| Flip-out | *hs-FLP,act >CD2>GAL4;UAS-Dcr2,UAS-GFP* | Lei Zhang |
| MARCM | *UAS-mCD8::GFP,hs-FLP;FRT42D,tub-GAL80; tub-GAL4* | Alex Gould |
| | *FRT42D* | |
| | *yw,hs-FLP;tub-GAL4,UAS-mCD8::GFP/CyO; FRT2A, tub-GAL80/TM6B* | |
| | *FRT2A* | |
| | *tub-GAL4,UAS-nlsGFP6xmycNLS,hs-FLP;FRT82B, tubP-GAL80 LL3/TM6B* | |
| | *FRT82B* | |
| | *FRT82B,pros17* | BL 5458 |
| GAL4 | *eyR16F10-GAL4* | BL 48737 |
| Reporter | *Dpn::GFP* | William Chia |
| | *gcm-lacZ* | BL 5445 |
| | *Tll::EGFP* | BL 30874 |
| | *Pros::GFP* | VDRC 318463 |
| RNAi | *UAS-mCherry RNAi* | BL 35785 |
| | *UAS-notch RNAi* | VDRC 100002 |
| | *UAS-pros RNAi* | BL 26745 |
| | *UAS-gcm RNAi* | VDRC 110539 |
| | *UAS-med12 RNAi* | BL 34588 |
| | *UAS-ND75 RNAi* | VDRC 100733 |
| | *UAS-EcR RNAi* | VDRC 37058 |
| | *UAS-Atg1 RNAi* | BL 26731 |
| | *UAS-tll RNAi* | VDRC 330031 |

*Continued on next page*

*Continued*

| System | Genotype | Source |
|---|---|---|
| Overexpression | *UAS-l'sc* | BL 51670 |
| | *UAS-N^ACT^* | Helena Richardson |
| | *UAS-p35* | BL 5072 |
| | *UAS-lacZ* | BL 51670 |
| | *UAS-pros* | Andrea Brand |
| | *UAS-gcm* | BL 5446 |
| | *UAS-dp110^CAAX^* | BL 25908 |
| | *UAS-myc* | BL 9674 |
| | *UAS-RedStinger* | BL 8547 |
| | *UAS-EcR^DN^* | BL 9451 |
| | *UAS-tll* | Claude Desplan |
| | *UAS-hth* | Kieran Harvey |
| | *UAS-D* | BL 8861 |

Fly stocks were reared on standard media at 25 °C. For larval dissection, brains were dissected at wandering L3 stages (approximately 120 hr ALH). For pupal dissection, white pupae were selected and allowed to age to the desired stages.

Knockdown and overexpression experiments using *eyR16F10-GAL4* were moved to 29 °C after overnight embryo collection until dissection. Experiments using *tub-GAL80^ts^;eyR16F10-GAL4,* embryos were reared at 25 °C for 1 day, larvae were then move to repressive 18 °C for 5 days to inhibit GAL4 activity, and transferred to permissive 29 °C to induce GAL4-mediated transgene expression for 2 days prior to dissection. Knockdown and overexpression flip-out clones were induced at 24 hr ALH at 37 °C for 10 min and then allowed to develop at 29 °C until dissection except for *pros RNAi* clones that were let developed at 25 °C to allow a mild knockdown of *pros*. *FRT42D* and *FRT42D;UAS-tll* MARCM clones were induced at 24 hr ALH (*Figure 2A–D*) or 48 h ALH (*Figure 5—figure supplement 2A–B*) at 37 °C for 15 min. *FRT82B* and *FRT82B,pros^17^* MARCM clones were induced at 24 hr ALH at 37 °C for 8 min.

## Immunostaining

Larval and pupal brains were dissected in phosphate buffered saline (PBS), fixed in 4% formaldehyde for 20 min and rinsed three time in 0.3% or 0.5% PBST (PBS +0.5% Triton), respectively at room temperature. For immunostaining, brains were incubated in primary antibodies overnight at 4 °C, followed by two washes in 0.3–0.5% PBST and then an overnight secondary antibody incubation at 4 °C or 3 hr at room temperature. Brains were afterwards rinsed two times in 0.3–0.5% PBST and incubated in 50% glycerol in PBS for 20 min. Samples were mounted in 80% glycerol in PBS, on glass slides and sealed with coverslips (22x22 mm, No.1.5, Knittel) for image acquisition. The primary antibodies used were: rat anti-Dpn (1:200, Abcam 195172), chick anti-GFP (1:1000, Abcam 13970), mouse anti-EcR^common^ (1:50, DSHB Ag10.2), rabbit anti-Dcp-1 (1:100, Cell Signalling 95785), rat anti-Mira (1:100, Abcam 197788), rat anti-Pros (a gift from Fumio Matsuzaki), rabbit anti-ß-Galactosidase (1:200, a gift from Helena Richardson), rabbit anti-Ase (a gift from Lily Jan and Yuh Nung Jan), mouse anti-Repo (1:50, DSHB 8D12), rabbit anti-PatJ (1:500, a gift from Helena Richardson), anti-Ey (1:50, DSHB), guinea pig anti-Slp (1:200, a gift from Kuniaki Saito), rabbit anti-Tll (1:200, a gift from Kuniaki Saito). Secondary donkey antibodies conjugated to Alexa 555 and Alexa 647, and goat antibodies conjugated to Alexa 405, 488, 555, and 647 (Molecular Probes) were used at 1:500.

## Image acquisition and processing

Images were acquired using Olympus FV3000 confocal microscope with 40 x (NA 0.95, UPLSAPO) and 60 x (NA 1.30, UPLSAPO) objectives. Δz=1.5 μm. Images were processed using Fiji (https://imagej.net/Fiji).

Quantification was performed in Fiji or via 3D reconstruction in Imaris (Bitplane). To quantify NB numbers (*Figures 1C and 4E*, *Figure 1—figure supplements 1N and 2E*, *Figure 3—figure supplement 1G*), the numbers of Dpn$^+$ cells in the OPC were measured. To quantify the nuclear sizes of the medulla NBs (*Figure 1E*), nuclear diameters identified by Dpn were measured. To quantify NB and NE volumes in the OPC (*Figure 2A*), the Dpn$^+$ and PatJ$^+$ volumes were measured for the NBs and the NE, respectively. To quantify the ratio of NBs (*Figure 2J, M*, *Figure 3P*, *Figure 4I*, *Figure 5I, P*, *Figure 1—figure supplement 1E*, *Figure 5—figure supplement 2I*) or glia (*Figure 5D*) in clones, GFP$^+$ clones were first detected and GFP$^+$ volumes were measured. The GFP$^+$ areas were then utilised to make a mask for Dpn$^+$ or Repo$^+$ cells within the clones, which was in turn used to measure NB or glial cell numbers, respectively. The NB or glial ratio in clones is calculated as the ratio of the number of GFP$^+$Dpn$^+$ or GFP$^+$Repo$^+$ cells over the GFP$^+$ volume. To quantify the distributions of apoptotic NBs expressing Ey, Slp, and Tll (hereby referred as tTF; *Figure 3K*), Dpn::GFP$^+$Dcp-1$^+$ cells were first identified and used as a mask for cells that are positive for tTFs. The ratio of each tTF is calculated as the ratio of the number of GFP$^+$Dcp-1$^+$tTF$^+$ cells over the number of GFP$^+$Dcp-1$^+$ cells. To quantify the number of glia in the medulla (*Figure 5—figure supplement 2H*), the total numbers of Repo$^+$ cells in the deep section of the medulla were measured. Images were assembled in Affinity Publisher 2. Schematics were created in Affinity Design 2. Scale bars = 10 μm or 50 μm per indicated in figures.

## Statistical analyses

The biological replicates in our experiments are represented by a minimum of three animals per genotype. The n number reflects brain lobe numbers unless otherwise specified. Statistical analyses were performed, and graphs were plotted in GraphPad Prism 9. In graphs, data is represented as mean ± standard error of the mean (SEM). For comparisons between two conditions, P-values were calculated by non-parametric Mann-Whitney tests when data if not normally distributed. For comparisons between more than two experimental conditions, p-values were calculated by ordinary one-way ANOVA tests for normally distributed data or data with small sampling size, that is n<5 (*Figures 1C and 2E*). For non-normally distributed data, Kruskal-Wallis tests were employed. Holm-Šídák's or Dunn's tests were used to correct for multiple comparisons following one-way ANOVA and Kruskal-Wallis tests, respectively. For the comparisons of apoptotic NBs expressing different temporal factors at 12–16 h APF (*Figure 3K*), ordinary two-way ANOVA test was used, followed by Šídák's test for multiple comparisons. (*) $p<0.05$ (**) $p<0.01$, (***) $p<0.001$, (****) $p<0.0001$, (ns) $p>0.0$. If quantifications were not performed, the number of samples in which specific phenotypes were observed, is mentioned in the text.

## Acknowledgements

We are grateful to Lei Zhang, Alex Gould, Claude Desplan, Fumio Matsuzaki, Helena Richardson, Kuniaki Saito, Yuh Nung Jan for generous sharing of fly stocks and antibodies. We would also like to thank Holger Apitz, Édel Alvarez-Ochoa, and Qian Dong for fruitful discussions and critical reading of the manuscript. We also thank Bloomington *Drosophila* Stock Centre, Vienna *Drosophila* Resource Centre, and Developmental Studies Hybridoma Bank for fly stocks and antibodies. We would like to also thank OZDros for *Drosophila* quarantine, Peter MacCallum Cancer Institute CAHM for microscopy assistance. PKN is funded by a PhD scholarship from the Department of Anatomy and Physiology, The University of Melbourne and the Vingroup Science and Technology Scholarship Program for Overseas Study for Master's and Doctoral Degrees. LYC's laboratory is supported by funding from the Peter MacCallum Cancer Foundation.

## Additional information

### Funding

| Funder | Grant reference number | Author |
|---|---|---|
| University of Melbourne | Graduate student scholarship | Phuong-Khanh Nguyen |
| Peter MacCallum Foundation | Strategic Support for Research Leaders | Louise Y Cheng |

The funders had no role in study design, data collection and interpretation, or the decision to submit the work for publication.

### Author contributions

Phuong-Khanh Nguyen, Conceptualization, Data curation; Louise Y Cheng, Conceptualization, Data curation, Writing – original draft

### Author ORCIDs

Phuong-Khanh Nguyen ⓘ https://orcid.org/0000-0001-8440-5360
Louise Y Cheng ⓘ https://orcid.org/0000-0001-9712-4082

### Decision letter and Author response

Decision letter https://doi.org/10.7554/eLife.96876.sa1
Author response https://doi.org/10.7554/eLife.96876.sa2

## Additional files

### Supplementary files

• MDAR checklist

### Data availability

We have uploaded source data to Dryad (https://doi.org/10.5061/dryad.tmpg4f56p).

The following dataset was generated:

| Author(s) | Year | Dataset title | Dataset URL | Database and Identifier |
|---|---|---|---|---|
| Nguyen P | 2024 | *Drosophila* medulla neuroblast termination via apoptosis, differentiation and gliogenic switch is scheduled by the depletion of the neuroepithelial stem cell pool | https://doi.org/10.5061/dryad.tmpg4f56p | Dryad Digital Repository, 10.5061/dryad.tmpg4f56p |

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
