## [Editor Report]

This study presents an important finding on the mechanisms by which medulla neural stem cells in *Drosophila* optic lobe, whose division mode largely resembles mammalian neural stem cells, terminate cell proliferation in neural development. The evidence supporting the conclusions of the authors is convincing. The work will be of interest to stem cell biologists and developmental biologists.

---

## [Decision Letter]

[Editors' note: this paper was reviewed by Review Commons.]

---

## [Author Response]

1. General Statements [optional]

We thank all the reviewers for their time and their constructive criticism, based on which we will revise our manuscript. All our responses are indicated in blue.

2. Description of the planned revisionsReviewer #1 (Evidence, reproducibility and clarity (Required)):The manuscript by Nguyen and Cheng is investigating the timing and mechanism of cessation of neuroblasts in the pupal optic lobe. Previous studies by several groups have determined the spatial and temporal factors required for the neuroepithelial to neuroblast transition and neuroblast to neural/glycogenesis in third instar larvae such that neuroblasts are eliminated. The mechanism of elimination of neuroblasts in the VNC or mushroom bodies have been investigated, but the mechanism(s) and the timing of elimination of medulla neuroblasts has not been investigated. The authors suggest that medulla neuroblasts are eliminated via a combination of mechanisms including apoptosis, prospero induced size symmetric terminal differentiation and a switch to gliogenesis by gcm expression. Expression of Tailless also was found to affect the timing of medulla neuroblast termination. They also ruled out several mechanisms such as ecdysone pulses.Major commentsClearly written and logical flow to experiments and results not over interpreted.Clearly show that the neuroblast number and size decrease (12 to 18 hrs) and are eliminated by 30 hoursFigure 2a Marking of the Neuroepithelium. Would be more convincing if shown by PatJ expression and is clonal analysis. While the following panels use PatJ in clones suggesting are NE and NBs present it is more difficult to put into the context in the higher magnification images (Figure 2 D- M) and the Miranda expression in F' seems to be the entire lobe and it is not clear if would be any NE which does not agree with what is shown in panel A.

We will perform clonal analysis using MARCM to show that the elimination of medulla NBs (marked by Dpn) is accompanied by the depletion of the NE (marked by PatJ). For Figure 2 D, E, I, L, we will change the images from high magnifications to the whole lobes to clearly show the shift in the NE-NB transition upon Notch OE/KD.

Is difficult to see the neuroblasts in Figure 2 D D" and E. The figure does not match what is stated in the results in that the neuroblasts are difficult to observe. If the point is that there is fewer NE cells and more neuroblasts then this is hard to see. It has been previously shown that with Notch RNAi clones prematurely extrude form the NE (Egger 20210; Keegan 2023) and could be expressing more Neuroblast markers but this is not visible in the panels as shown. Are the images single focal plane or maximum projections? Imaging more deeply in the brain or viewing in cross section would account for these possibilities. The possibility that are more neuroblasts but not all at the surface of the OL should be addressed as this could also alter the overall results.Figure 2 is key to first point of the paper so needs to be addressed.

The images are single focal plane of the superficial layer of the medulla. We will specify this information in the figure legends. In addition, we will include cross-section images of the *notch RNAi* clones in our supplementary figure to show the delamination of precocious NBs and comment on this observation in the text.

Minor commentsWhy express volume of DPN in clone volume. Would make the point more clear and more strong be to express as number of NB in the 3-D volume of the clone. This measurement occurs in several figures.

We will redo the quantification as suggested.

Use of Miranda to mark NBs is unclear in Figure 2. Perhaps more clear in B and W.

We will redo the staining with Dpn instead of Mira to mark medulla NBs. Figures will be presented in B and W as suggested.

Make clear in figures (or figure legend) if single focal plane or projections.

We will make these suggested changes.

It is unclear what percentage of NB the Gal4 line eyR16F10 are expressed in. Veen 2023 state that the GAL4 is also expressed in neurons and at different levels whether deeper within the brain or superficially on the surface of the brain. At 16 APF it is expressed but it is not clear whether it is in all cells at a low level or only within a few cells

We will further characterize the expression of *eyR16F10-GAL4* in the pupal medulla as suggested.

Some RNAi lines referenced as previously validated and other are not. For example: EcR, Oxphos, Med27, Notch need references or confirmation of specificity to the intended target (qRT)

We will perform RT-qPCR to validate the knockdown efficiency of *UAS-med27 RNAi*. For RNAi stocks such as *UAS-EcR RNAi, UAS-Atg1 RNAi, UAS-notch RNAi* that have been previously used in other publications, we will provide appropriate references.

At least 2 animals per genotype were used. While I appreciate the technical difficulty of working in pupae this seems a bit low in terms of number of samples and data would be more robust with more numbers.

Any experiments in which less than 3 animals were used, we will redo the experiments and change our Materials and methods section accordingly.

Reviewer #1 (Significance (Required)):This provides mechanism and timing for the elimination of neuroblasts (NE to NB) that arise from the medulla. As these are most similar to mammalian brain development (Radial glial to NSC) this information provides more context to interpret the formation of glial and neurons in the adult optic lobe given the effect on timing and mechanisms of elimination.This paper would be of interest to developmental biologist who work with *Drosophila* or mice who are looking at neural development. An understanding of how neural diversity is achieved and the mechanisms behind this that can be dysfunctional in terms of etiology of neural diseases. Is a well done study for the most part that would be improved by clarifying some data and provided more replicates for robustness of the data.I am a developmental biologist working with *Drosophila* in larval and adult neural development.Reviewer #2 (Evidence, reproducibility and clarity (Required)):Lineages of neural stem cells are of great interest to understand how many neural types are generated. They produce very diverse neurons, often in a highly stereotyped series. However, they must terminate their life when the animal becomes functional or if neurons need time to become mature before birth.In the *Drosophila* optic lobes, neural stem cells are produced over a period of several days by a wave of neurogenesis that transforms a neuroepithelium into neural stem cells that undergo a series of temporal patterning steps. It has been reported that they finish their life when a symmetric division generates glial cells. The authors however analyze the end of a particular lineage, that of the latest born neural stem cells of the medulla.The paper shows that neural stem cells stop being produced when the neuroepithelium is consumed. But how do the latest born neural stem cells stop their lineage?The results show that they do so by several means, which is quite unexpected: they may die from apoptosis, or autophagy, by becoming glioblasts or by a terminal symmetric division.There are no major issues affecting the conclusions– The paper shows that the end of production of neural stem cells occurs the neuroepithelium is completely transformed. The experiments performed by the authors are fine and show that, if the transition is delayed, neural stem cells terminate their life later, and vice versa. However, the lifespan of the neural stem cells is not affected by the timing of the transition. Therefore, these experiments do not tell us how neural stem cells terminate their life, which is the central question of the study. The discussion should be written accordingly and the title and the model in Figure 6 modified to reflect the importance of the end of life of the stem cells, the main theme of the paper.

We agree that our said experiments did not elucidate how NBs terminate at the end of neurogenesis in the medulla. Nevertheless, our Figure 2 showed that the timing of NB termination in the medulla is dependent on the timing of the NE-NB transition (i.e. if born early, will terminate early, and if born late, will terminate late).

In Supplementary Figure 1, we showed that factors previously shown to be involved in NB termination in other lineages did not play similar roles in the medulla NBs. Thus, we think that NB termination in the medulla is likely regulated at the levels of the NE, but not the NBs themselves. Although we have briefly mentioned this in the discussion, we hope that by conducting the experiments suggested by the reviewer (see below), we can subsequently modify our model in Figure 6 and our discussion.

– The authors talk about Pros-dependent symmetric division and gliogenic switch as two separate processes, but these may be two sides of the same phenomenon. Tll+ gcm+ neural stem cells undergo Pros-dependent cell cycle exit, generating glial progeny. If the authors agree with this, could they update their model (and discussion) to reflect the fact that gliogenic switch occurs via a Pros-dependent symmetric division, and these are not two separate processes independently contributing to the depletion of the neural stem cell pool? Ideally, a triple staining between Dpn, Pros, and gcm would show that the symmetrically dividing cells seen by the authors are committed to the glial fate.

We agree it is likely that the two mechanisms are linked. In Figure 5C, we did show the triple staining the reviewer suggested: at 12h APF, there are Dpn+ NBs in the medulla that expressed both Pros::GFP and *gcm-lacZ*, suggesting that NBs likely undergo Pros-dependent cell cycle exit to generate glial progeny (Figure 5K). We will update our final model as suggested.

To investigate this further, we propose to test if gliogenesis is affected in *pros RNAi* clones. The results may shed light on whether Pros-mediated symmetric division is required for Gcm-mediated gliogenesis in the medulla.

– Why were Notch RNAi experiments assessed for the presence of neural stem cells at P12 and gcm RNAi experiments at P24? Given that most optic lobe neural stem cells disappear between P12-18, a subtle effect of gcm RNAi may have been missed. Do the authors have data for gcm RNAi at P12?

We hypothesized that the timing of NE-NB transition affects the timing of NB termination in the medulla. Because Notch KD was previously shown to induce precocious NE-NB transition in the OL i.e., NBs are born prematurely, we expected that this manipulation will lead to a premature elimination of the NBs.

In contrast, *gcm RNAi* which inhibits the switch into the glial cell fate of the NBs, is expected to prolong the neurogenic phase of the NBs, and thereby, their persistence by 24h APF when WT NBs are eliminated. We agree that there might be some subtle effects of *gcm RNAi* at 12h APF. However, we can show the number of NBs in *gcm RNAi* clones in the medulla at 12h APF as suggested.

– The authors should acknowledge that the inhibition of either apoptosis or autophagy alone may not be fully sufficient to prevent the death of NBs. In mushroom body neural stem cells, both processes must be inhibited simultaneously to produce a strong effect on their survival (Pahl et al. 2019, PMID 30773368).

We will add this information in our discussion.

– There is an important missing point that should be addressed: is there a specific point in time when all neural stem cells must stop their lineage wherever they are in the temporal series and either die or divide symmetrically? One possibility that is not discussed is that most neural stem cells end their life through a gliogenic symmetric division while those that were generated late must stop en route and die by apoptosis and/or autophagy. This would solve the strange diversity of end-of-life, which could be easily addressed by identifying the temporal stage of the neural stem cells that undergo apoptosis

We agree that it would be of interest to understand how there are diverse mechanisms by which medulla NBs terminate during pupal development. We agree with the above possibility that the reviewer raised such that most old NBs appear terminate through a gliogenic switch whereas newly generated NBs terminate en route via apoptosis. To investigate this, we will characterise the expression of some temporal transcription factors (e.g., Ey, Slp, Tll) in the NBs undergoing apoptosis at 12h APF as suggested.

Minor suggestions:

We agree with these minor suggestions and will amend these as suggested.

– Line 46: Specify that there are 8 type II neural stem cells in each hemisphere*.– The statement in lines 181-182 that "cell death, and not autophagy, makes a minor contribution to…" should be replaced with "apoptosis, and not autophagy," as autophagy is also a type of cell death.– The authors should adjust the logic of the section "Medulla neuroblasts terminate during early pupal development": Describe the wild-type pattern first (the decrease in the number of neural stem cells and their size with age) and then describe the perturbations aimed at disrupting the number and the size of neural stem cells– Line 151 should refer to Figure 2I-K, not Figure 2J-K.Referees cross-commentingHow can NBs die by different mechanisms?? This might only happen is they are in a different states, an issue that is not addressed.It has been shown that optic lobe NBs end their life by a symmetric, gliogenic last division at the end of the last temporal window, and not by PCD.It is likely, and the authors do hint at it, that NBs only die by PCD when they prematurely interrupt the temporal series in early pupation when neurons synchronously start undergoing maturation.I believe that the authors should explain this, if this is indeed their model, and show that NBs die while still in early temporal windows.Reviewer #2 (Significance (Required)):Lineages of neural stem cells are of great interest to understand how many neural types are generated. They produce very diverse neurons, often in a highly stereotyped series. However, they must terminate their life when the animal becomes functional or if neurons need time to become mature before birth.In the *Drosophila* optic lobes, neural stem cells are produced over a period of several days by a wave of neurogenesis that transforms a neuroepithelium into neural stem cells that undergo a series of temporal patterning steps. It has been reported that they finish their life when a symmetric division generates glial cells. The authors however analyze the end of a particular lineage, that of the latest born neural stem cells of the medulla.The paper shows that neural stem cells stop being produced when the neuroepithelium is consumed. But how do the latest born neural stem cells stop their lineage?The results show that they do so by several means, which is quite unexpected: they may die from apoptosis, or autophagy, by becoming glioblasts or by a terminal symmetric division.There are no major issues affecting the conclusionsReviewer #3 (Evidence, reproducibility and clarity (Required)):SummaryIn this manuscript, the authors address the timing and mechanisms responsible for the termination of medulla neuroblasts in *Drosophila* visual processing centres, also known as optic lobes. Through time course experiments the authors demonstrate the medulla NBs are completely eliminated by 30h APF during early pupal development. By manipulating the Notch signalling pathway as well as proneural genes such as lethal of scute, the authors show that altering the NE-NB transition is sufficient to change the timing of NB termination. In contrast, ecdysone signalling and components of the mediator complex, known to terminate proliferation of central brain NBs, are not required for the termination of medulla NBs. Medulla NBs sequentially express a variety of temporal transcription factors to promote cellular diversity, however, the authors demonstrate that altering temporal factors such as Ey, Sco or Hth, does not affect the timing of the medulla NBs termination. Interestingly however overexpression of the transcription factor tailless can cease medulla NB termination via the conversion of type I to type II NB fate. They further go on to show the importance of the differentiation factor, Prospero, in promoting the differentiation of medulla NBs as well as terminating medulla neurogenesis during pupal development. Finally, in addition to differentiation, the authors show another mechanism responsible for the cessation of neurogenesis which is the commencement of gliogenesis. Through manipulation of the neurogenic to gliogenic switch by knockdown or overexpressing the glial regulatory gene, gcm, the authors show that even though the downregulation of gcm is is not sufficient to induce NB persistence, gcm overexpression can cause premature termination of NBs.Major comments:– Are the key conclusions convincing?Yes, the key conclusions are convincing with proper controls, quantifications and statistical analyses.– Should the authors qualify some of their claims as preliminary or speculative, or remove them altogether?The conclusion that temporal transcription factors (TTF) do not affect the timing of medulla NB termination is somewhat preliminary. The authors investigated a simplified temporal series including Homothorax, Eyeless, Sloppy-paired, Dichaete and Tailless. However, there are additional temporal factors that have not been examined for their potential involvement in medullar NB termination. Previous reports have identified several other temporal factors that play a role in medulla TTF cascade, such as, SoxNeuro (SoxN) and doublesex-Mab related 99B (Dmrt99B) that start their expression in the NE similar to Hth, however, Dmrt99B is likely to be repressed much later than Hth (Li, Erclik et al. 2013, Zhu, Zhao et al. 2022). At this point, it remains challenging to completely rule out the possibility that other temporal factors play a role in medullar NB termination or have redundant functions in regulating the timing of medulla NB cessation. It is suggested to tone down this claim and provide a brief discussion on alternative possibilities, citing relevant papers on the functions of other temporal factors in medullar NBs.

We agree and will make the suggested changes.

– Would additional experiments be essential to support the claims of the paper? Request additional experiments only where necessary for the paper as it is, and do not ask authors to open new lines of experimentation.Loss of pros by RNAi caused the formation of ectopic NBs and the NBs persist even at 24h APF. Do these NBs persist at 30h or 48h APF? Does overexpression of Pros result in early termination of medulla NBs?

We will do these experiments in clones as suggested.

– Are the suggested experiments realistic in terms of time and resources? It would help if you could add an estimated cost and time investment for substantial experiments.Yes, I believe the suggested experiments are realistic in terms of time and resources, with an estimation of 3 months to complete the experiments.– Are the data and the methods presented in such a way that they can be reproduced?Yes.– Are the experiments adequately replicated and statistical analysis adequate?The experiments are straight forward and were performed with proper controls, supported by quantifications and proper statistical analyses. However, there is no mention about how many replicates were used.

We will add this information in our Material and Methods section.

Minor comments:1. The authors use the eyR6F10-Gal4 driver in certain experiments. The eyR6F10-Gal4 driver is however expressed only in a subset of medulla NBs. Can the authors comment on what percentage of medulla NBs is the driver expressed in?

We will further characterize *eyR16F10-GAL4* as suggested.

2. Does the EGFR signalling pathway or JAK/STAT pathway affect the timing of termination of medulla NBs? Experiments are not necessary. The author can speculate on their roles.

We will modify our discussion accordingly.

3. Figure 1C has a p value of only 0.03 (*) but shows a strong reduction in the number of Dpn+ cells from 12h to 18h, etc. Is this correct? Also, is the p value the same for the comparison between 12h and 24h as well as 12h and 30h APF?

Yes. P-values showed no significant differences between 18-24h and 24-30h APF.

4. The controls in figure 2B and to some extent figure 2H show one major outlier (much higher than the other brain lobes in the control). Will the removal of this outlier affect the significance/ p-value of the experiment?

Removing the outlier in 2B do not change the statical results. Removing the outlier in figure 2H changes the P-value from **p=0.004 to *p=0.010. Overall, we do not think that outliers drastically change our conclusions.

5. In figure 2B what is the p-value between 12h and 18h APF? Is it *** as well?

No, it’s not significant (likely due to the outlier at the 12h APF timepoint)

6. Line 84 of the introduction introduces Tll, Gcm and Pros for the first time in the manuscript and should be written out in full.

We will change this.

- Are prior studies referenced appropriately?Yes.- Are the text and figures clear and accurate?Yes.- Do you have suggestions that would help the authors improve the presentation of their data and conclusions?Quite a few of data mentioned in the manuscript have been described as data not shown. I think it would be nice to show quantifications or representative images in the supplementary figures.

We will add some data mentioned in the manuscript such as Notch OE clones at later pupal stages (Figure 2), Myc OE (Supplementary Figure 1), and manipulations of the temporal factors (Supplementary Figure)

Reviewer #3 (Significance (Required)):Since the mechanisms by which medulla NBs are terminated are currently unknow, this is an important and interesting study to understand how medulla neuroblasts in the optic lobe are terminated. The balance between stem cell maintenance and differentiation is critical for proper brain development and the results presented in this paper are impactful. Furthermore, *Drosophila melanogaster* is an excellent model to study stem cell niches and neuroblast temporal patterning. The authors provide key mechanisms namely cell death, Pros-mediated differentiation and the gliogenic switch that contribute to a better understanding of how the NB progenitor pool can be terminated in the Drosophila OL, which is largely supported by the data.- Place the work in the context of the existing literature (provide references, where appropriate).So far, most work in this field has focused on the regulation of the temporal factors to promote the progression of the TTF transcriptional cascade and thereby diversity of the neural progenitors (Li, Erclik et al. 2013, Naidu, Zhang et al. 2020, Ray and Li 2022, Zhu, Zhao et al. 2022). Furthermore, work on pathways such as EGFR and Notch signalling that allows the proneural wave to progress and subsequently induce neuroblast formation in a precise and orderly manner have also been studied (Yasugi, Umetsu et al. 2008, Yasugi, Sugie et al. 2010). Here, considering previous literature, the authors move one step forward to determine how and when these neuroblast progenitors cease proliferation during development thus providing mechanisms for the regulation of the neuroepithelial stem cell pool, its timely conversion into NSCs and the switch from neurogenesis to gliogenesis thus providing important implications for brain size determination and function.- State what audience might be interested in and influenced by the reported findings.Stem cell research, neurobiologists and developmental biologists.- Define your field of expertiseStem cells, developmental biology

List of planned experiments for revision:

Lineage analysis (MARCM) of the medulla NE cells (PatJ+) and NBs (Dpn+) from 12-30h APF.Re-staning of *mCherry RNAi*, *notch RNAi*, and *N^ACT^* clones with PatJ and Dpn at larval and pupal stages.*pros RNAi* clones at 48-72h APF to test the persistence of medulla NBs.*pros* overexpression clones at 12h APF to test for precocious termination of medulla NBs.*gcm RNAi* clones at 12h APF to look for subtle effects of gliogenic inhibition in the medulla.Characterisation of temporal factor expressions in apoptotic NBs in the wildtype medulla at 12h APF.Characterisation of *eyR16F1-GAL4* expression in the medulla at 12h APF.RT-qPCR to validate *UAS-med27 RNAi*.